# Oncostatin M Is Related to Polycystic Ovary Syndrome-Case Control Study

**DOI:** 10.3390/biomedicines12020355

**Published:** 2024-02-02

**Authors:** Figen Efe Camili, Merve Akis, Ertan Adali, Adnan Adil Hismiogullari, Mine Islimye Taskin, Gurhan Guney, Selim Afsar

**Affiliations:** 1Department of Obstetrics and Gynecology, School of Medicine, Balikesir University, 10145 Balikesir, Türkiye; efefigenefe@gmail.com (F.E.C.); ertanadali@gmail.com (E.A.); minetaskin1302@yahoo.com.tr (M.I.T.); gurhanguney@yahoo.com (G.G.); 2Department of Medical Biochemistry, School of Medicine, Balikesir University, 10145 Balikesir, Türkiye; merve.akis@balikesir.edu.tr (M.A.); ahismiogullari@gmail.com (A.A.H.)

**Keywords:** polycystic ovary syndrome, oncostatin M, adipokines, C-reactive protein, inflammation

## Abstract

**Background:** Oncostatin M, a novel adipokine, plays a role in oogenesis, lipogenesis, and inflammation and may contribute to polycystic ovary syndrome pathogenesis and related metabolic problems. Adipokines are believed to contribute to developing polycystic ovary syndrome and its accompanying metabolic parameters, such as dyslipidemia, insulin resistance, and cardiovascular diseases. **Methods:** In this case–control study, the patients were grouped in a 1:1 ratio into either the polycystic ovary syndrome (*n =* 32) or the control group (*n =* 32). Serum levels of fasting glucose, insulin, C-reactive protein, low-density lipoprotein cholesterol, high-density lipoprotein cholesterol, triglyceride, white blood cell count, thyroid-stimulating hormone, luteinizing hormone, follicle-stimulating hormone, total testosterone, prolactin, estradiol, homeostasis model assessment of insulin resistance, and oncostatin M were analyzed. **Results:** Oncostatin M levels were significantly lower, but C-reactive protein levels were substantially higher in the polycystic ovary syndrome group than in the control group (*p* = 0.002, *p* = 0.001, respectively). Oncostatin M was inversely correlated with total cholesterol, non-high-density lipoprotein cholesterol, fasting glucose, and the luteinizing hormone/follicle-stimulating hormone ratio (ρ = −0.329, *p* =0.017; ρ = −0.386, *p* = 0.005; ρ = −0.440, *p* = 0.001; ρ = −0.316, *p* = 0.023, respectively). Conversely, there was no correlation between oncostatin M and total testosterone level (ρ = 0.220; *p* = 0.118). In the context of inflammation and metabolic parameters, oncostatin M was inversely correlated with C-reactive protein, homeostatic model assessment for insulin resistance score, and low-density lipoprotein cholesterol (ρ = −0.353, *p* = 0.019; ρ = −0.275, *p* = 0.048; ρ = −0.470, *p* < 0.001, respectively). **Conclusions:** Plasma oncostatin M levels were considerably lower in patients with polycystic ovary syndrome than in the control group, and this was inversely correlated with the hormonal and metabolic parameters of polycystic ovary syndrome. Thus, oncostatin M may be a novel therapeutic target for polycystic ovary syndrome and its metabolic parameters.

## 1. Introduction

Polycystic ovary syndrome is a common reproductive and endocrinological disorder that affects 6–10 percent of women and is characterized by ovulatory dysfunction (chronic anovulation), polycystic ovarian morphology, and biochemical and/or clinical hyperandrogenism when two out of three Rotterdam criteria’s (2003) are present. It is important to note that the diagnosis of polycystic ovary syndrome is established only after alternative causes of androgen excess have been ruled out, including hyperprolactinemia, non-classical congenital adrenal hyperplasia, androgen-secreting ovarian or adrenal tumors, hypothyroidism, Cushing’s syndrome, and acromegaly. Polycystic ovary syndrome is typically identified when a patient experiences severe reductions in their quality of life due to hair loss, alopecia, acne, and problems related to infertility. Additionally, metabolic parameters, including dyslipidemia, endothelial dysfunction, atherosclerosis, insulin resistance, obesity, fatty liver disease, coagulation disorders, and increased risk of cardiovascular disease, are associated with polycystic ovary syndrome [1,2]. Although its pathogenesis remains unclear, a significant amount of data shows that adipokines may play a role in the etiology of polycystic ovary syndrome [3,4]. In addition to serving as the body’s largest endocrine organ, adipose tissue releases a myriad of adipokines that are involved in numerous pathological processes, including reproduction, inflammation, glucose and lipid metabolism, regulation of energy metabolism, and insulin resistance [5].

Oncostatin M is a novel adipokine and a member of the IL-6 cytokine family that stimulates the Janus kinase/signal transducer and activator of the transcription pathway by binding to a transmembrane receptor. Oncostatin M has many biological functions, including lipogenesis/adipogenesis, hematopoiesis, osteogenesis, and inflammation regulation. On the other hand, chronic inflammatory diseases related to fibrosis, such as rheumatoid arthritis, inflammatory bowel diseases, systemic sclerosis, and cirrhosis, have high levels of oncostatin M [6].

It has also been shown that human oocytes and granulosa cells express oncostatin M and its receptor. Additionally, it has been reported that oncostatin M has stimulatory effects on the number and growth of primordial germ cells in the ovaries [7]. Oncostatin M may also stimulate the production of additional growth factors that support and facilitate the development of primordial follicles. The increase in oncostatin M signaling following human chorionic gonadotropin administration and subsequent ovulation indicates that oncostatin M signaling plays a fundamental role in ovulation [8].

Therefore, in light of the current literature, we hypothesized that oncostatin M might contribute to the pathophysiology of polycystic ovary syndrome because of its relationship with ovulation, insulin resistance, and inflammation.

## 2. Materials and Methods

### 2.1. Participants

Participants were recruited for the study after informed consent was obtained between July 2021 and July 2022. The participants were allocated in this study on a 1:1 ratio as polycystic ovary syndrome group (*n =* 32) or control group (*n =* 32). This study included women between 20 and 35 years with a body mass index ranging from 18 to 30. Body mass index (BMI) was calculated by dividing the patient’s weight in kilograms by the square of height in meters. The Rotterdam criteria were used to identify patients with polycystic ovary syndrome for this research [9]. Polycystic ovary syndrome group consisted of phenotype-A patients [10]. The control group consisted of women with regular menstrual periods, normal ovaries on ultrasonographic evaluation, normal hormonal status, and normal homeostasis model assessment of insulin resistance values.

### 2.2. Biochemical Analysis

Blood samples were collected between the third and fifth days of the menstrual cycle after at least eight hours of fasting, and anthropometric measurements were completed. After centrifugation at 1500× *g* for 10 min, the serum of patients was halved, and the first part was stored at −80 °C for further analysis of oncostatin M, then it was used without diluting. The second part of the serum was used to measure the level of fasting glucose, insulin, white blood cell, C-reactive protein, high-density lipoprotein, low-density lipoprotein, triglycerides, thyroid-stimulating hormone, luteinizing hormone, follicle-stimulating hormone, total testosterone, prolactin, and estradiol. Homeostasis model assessment of insulin resistance value was calculated using the following formula: fasting insulin (μU/L) × fasting glucose (mmol/L)/405. If the score was greater than 2.4, the patient was considered insulin-resistant. Non-high-density lipoprotein cholesterol level was calculated using the formula: total cholesterol − high-density lipoprotein cholesterol. Plasma levels of oncostatin M were measured with an enzyme-linked immunoassay kit (E-EL-H2247, Elabscience, Houston, TX, USA) that had a sensitivity of 9.38 pg/mL and a detection rate of 15.63–1000 pg/mL.

### 2.3. Statistical Analysis

Statistical and power analyses were performed using the open-source Jamovi (version 2.3.21) statistical software program and G*Power software (version 3.1), respectively. A few papers in the literature have explored oncostatin M; therefore, the minimum sample size was calculated as 28 per group based on an α error of 0.05, power of 0.90, and effect size of 0.7 [6,7,8,11,12]. An independent sample *t*-test and the Mann–Whitney U test were used to compare variables with normal and abnormal distributions, respectively. Partial correlation analyses were performed using either the Pearson or Spearman coefficients according to the variables with normal and abnormal distributions. A *p*-value < 0.05 was considered statistically significant.

## 3. Results

Blood samples were collected from 64 women (polycystic ovary syndrome group, *n =* 32; control group, *n =* 32), and no significant differences were found in body mass index, triglyceride, high-density lipoprotein cholesterol, white blood count, thyroid-stimulating hormone, prolactin, and estradiol levels between the study groups. Table 1 summarizes the study groups’ metabolic, anthropometric, and hormonal results.

Non-high-density lipoprotein cholesterol, low-density lipoprotein cholesterol, total testosterone, and luteinizing hormone/follicle-stimulating hormone ratio were considerably higher in the polycystic ovary syndrome group than in the control group (*p* < 0.001; Table 1). Oncostatin M levels were significantly lower, but C-reactive protein levels were substantially higher in the polycystic ovary syndrome group than in the control group (*p* = 0.002, *p* = 0.001, respectively; Table 1, Figure 1).

Regarding the association between oncostatin M and biochemical variables, oncostatin M was inversely correlated with fasting glucose, total cholesterol, non-high-density lipoprotein cholesterol, and luteinizing hormone/follicle-stimulating hormone ratio (ρ = −0.329, *p* = 0.017; ρ = −0.386, *p* = 0.005; ρ = −0.440, *p* = 0.001; ρ = −0.316, *p* = 0.023, respectively). Conversely, there was no correlation between oncostatin M and homeostasis model assessment of insulin resistance and total testosterone level, respectively (ρ = 0.275, *p* = 0.048; ρ = 0.220, *p* = 0.118). Table 2 summarizes the correlation between oncostatin M and biochemical markers.

In the context of the inflammation and metabolic parameters, oncostatin M was inversely correlated with C-reactive protein, homeostasis model assessment of insulin resistance, and low-density lipoprotein cholesterol (ρ = −0.353, *p* = 0.019; ρ = −0.275, *p* = 0.048; ρ = −0.470, *p* < 0.001, respectively; Table 2, Figure 2). The correlation of oncostatin M with C-reactive protein, homeostasis model assessment of insulin resistance, and low-density lipoprotein was plotted in Figure 2.

## 4. Discussion

It was shown in this study that oncostatin M levels were markedly lower in polycystic ovary syndrome patients compared to the control group. It was also discovered that patients with polycystic ovary syndrome had significantly higher levels of serum C-reactive protein, homeostasis model assessment of insulin resistance, and total testosterone compared to the control group. Additionally, a significant negative correlation was detected between oncostatin M levels and fasting plasma glucose, homeostasis model assessment of insulin resistance, C-reactive protein, non-high-density lipoprotein cholesterol, low-density lipoprotein cholesterol, total cholesterol values, and luteinizing hormone/follicle-stimulating hormone ratios.

In a recently published study by Nikanfar S et al. [8], patients with polycystic ovary syndrome were compared to healthy controls regarding the follicular fluid levels of oncostatin M and its receptor. It was found that oncostatin M and its receptor levels in follicle fluid were significantly lower in patients with polycystic ovary syndrome. In that study, Nikanfar et al. [8] attributed the decrease in oocyte maturation and the increase in the number of immature oocytes in patients with polycystic ovary syndrome to the increased expression of SOCS3. This increase in expression interfered with the levels of oncostatin M and its receptor in the follicle fluid. Additionally, it was shown that inflammatory cytokines mediate SOSC3 elevation in patients with polycystic ovary syndrome.

Regarding chronic low-grade inflammation in polycystic ovary syndrome, most of the articles in the literature have revolved around C-reactive protein, an acute-phase protein produced by the liver in response to interleukin-6 and tumor necrosis factor stimulation. Furthermore, C-reactive protein is secreted directly from adipose tissue. An increasing amount of evidence indicates that C-reactive protein is a reliable indicator of the inflammatory process within blood vessels and a crucial factor in the progression of cardiovascular disease [10]. Kelly et al. [13] were the first to report the link between elevated C-reactive protein levels and polycystic ovary syndrome. However, their study included only a small number of patients. Even after adjusting for age and body mass index, they detected a significant elevation in serum C-reactive protein levels. Multiple studies have shown that C-reactive protein levels are increased in patients with polycystic ovary syndrome [14,15]. A meta-analysis of Escobar-Moralle et al. [16] reported that women with polycystic ovary syndrome had higher C-reactive protein levels than control groups. A second meta-analysis by Toulis et al. [17] confirmed these results. Recent studies have also revealed a marked reduction in C-reactive protein levels with metformin treatment in patients with polycystic ovary syndrome [16]. There seems to be a positive association between C-reactive protein and insulin resistance, body weight, and total body fat mass. Nonetheless, whether the inflammation is caused by polycystic ovary syndrome or insulin resistance is unclear. Further studies are required to determine the underlying cause of increased C-reactive protein levels among women with polycystic ovary syndrome. Oncostatin M is a cytokine that belongs to the interleukin-6 family and can bind to two distinct receptors, the leukemia inhibitory factor receptor and the oncostatin M receptor, via a complex that includes the common glycoprotein 130 subunit [18]. It could be hypothesized that oncostatin M may also be involved in the inflammatory process because C-reactive protein is linked to interleukin-6. Likewise, oncostatin M is linked to interleukin-6. A substantial body of research supports the involvement of oncostatin M in the inflammatory process [19,20].

Although all inflammatory cytokines in this study were not examined, it was revealed that C-reactive protein levels were notably elevated in patients with polycystic ovary syndrome compared to the control group. It was also observed that plasma C-reactive protein levels were negatively correlated with the plasma levels of oncostatin M in patients with polycystic ovary syndrome. Based on elevated C-reactive protein levels, it could be concluded that patients with polycystic ovary syndrome [13] may have chronic low-grade inflammation. This inflammation could be associated with oncostatin M, similar to the findings of the study conducted by Nikanfar S et al. [8]. Another study from Elks CM et al. [21], which evaluated the relationship between oncostatin M and inflammation, demonstrated that cancellation of adipocyte oncostatin M signaling disrupts adipose tissue homeostasis, resulting in inflammation and insulin resistance without making significant changes in fat mass. In addition, Elks CM and colleagues [21] also observed that treatment with oncostatin M induced gene expression of Timp1, Igfbp3, and Spp1 while alleviating insulin resistance. Elks CM et al.’s [21] study provides solid proof for the relationship of oncostatin M with inflammation and insulin resistance. Similarly, the negative correlation between oncostatin M, homeostatic model assessment for insulin resistance, and C-reactive protein values was detected in our study.

In a different study published by Akarsu M and his colleagues [22], 51 patients with insulin resistance and 33 healthy controls without insulin resistance were compared regarding oncostatin M, homeostasis model assessment of insulin resistance, C-reactive protein, total cholesterol, low-density lipoprotein cholesterol, high-density lipoprotein cholesterol, triglyceride levels, and waist circumference. As a result, Akarsu M and colleagues [22] found that waist circumference, fasting glucose, insulin, C-reactive protein, high-density lipoprotein cholesterol, oncostatin M, and homeostasis model assessment of insulin resistance values were statistically significantly different in the patients with high insulin resistance. In that study, the researchers also noted a significant positive correlation between oncostatin M levels and both C-reactive protein and homeostasis model assessment of insulin resistance values. A negative correlation was found between oncostatin M levels and both homeostasis model assessments of insulin resistance and C-reactive protein levels in our study. Including groups of obese patients and a relatively small control group in Akarsu M and colleagues’ study may have resulted in different findings compared to our study. In their research, Akarsu M and colleagues suggested that oncostatin M may have contributed to the development of insulin resistance by interacting with specific adipokines, especially in obese patients. Additional studies are needed to investigate the relationship between oncostatin M, inflammatory markers, and insulin resistance to gain a better understanding of the role of oncostatin M in metabolism.

In addition to metabolic and hormonal factors, the neuroendocrine system plays an essential and central role in the pathophysiology of polycystic ovary syndrome and remains up-to-date in recent research. An imbalance in the pattern of gonadotropin-releasing hormone production can lead to a disruption in the hypothalamic-pituitary-ovarian or adrenal axis, which in turn has been associated with the development of polycystic ovary syndrome. This imbalance causes a higher release of luteinizing hormone compared to follicle-stimulating hormone. Hypothalamic gonadotropin-releasing hormone neurons are the primary connection between reproductive function and metabolic status. They also play a crucial role as the final output pathway for the central regulation of the reproductive axis [23].

In a study conducted by Reynolds MF and colleagues [24], it was found that valproic acid may be associated with polycystic ovary syndrome. In the same study, Reynolds MF and colleagues demonstrated that polycystic ovary syndrome is caused by impaired secretion of N-methyl-D-aspartate and impaired neuronal transmission of gamma-aminobutyric acid, which affects the secretion of gonadotropin-releasing hormone. In another study published by Igaz P and colleagues [25], it was shown that oncostatin M could regulate gonadotropin-releasing hormone secretion through N-methyl-D-aspartate secretion, which is a glutamate agonist. In line with previous research, it was observed that the ratio of luteinizing hormone to follicle-stimulating hormone was significantly higher in patients with polycystic ovary syndrome compared to the control group in our study. A significant negative correlation was also found between the ratio of luteinizing hormone to follicle-stimulating hormone and levels of oncostatin M. Taken together, it could be argued that oncostatin M may interact through the neuroendocrine system in polycystic ovary syndrome. More thorough studies on the relationship between oncostatin M and the secretion of gonadotropin-releasing hormone could help us better understand the role of oncostatin M in the development of polycystic ovary syndrome.

Patients with polycystic ovary syndrome were found to have higher levels of low-density lipoprotein and non-high-density lipoprotein cholesterol than the control group in our study. In addition, an inverse correlation between oncostatin M levels and elevated cholesterol levels was noted. Some previous in vitro research suggested that oncostatin M directly causes dyslipidemia and atherosclerosis, which contradicts our study. For example, in their study, Liu et al. [26] showed that oncostatin M plays a supportive role in ox-LDL-induced foam cell formation and inflammation. Therefore, silencing oncostatin M may be beneficial for treating atherosclerosis. In another study by Zhan et al., oncostatin M receptor-β deficiency in macrophages was responsible for inhibiting atherogenesis. Zhan et al. [27] also observed that despite a fat-rich diet, there was no difference in low-density and non-high-density lipoprotein cholesterol levels in mice, and these mice did not develop atherosclerosis. They attributed this effect to the inhibitory nature of the absence of oncostatin M receptors in the JAK2/STAT3 pathway. In our opinion, both in vitro studies should be validated by in vivo studies, as the interaction between statins and serum lipids may differ in the human body [28].

Various studies have also revealed that higher serum testosterone levels accompanying polycystic ovary syndrome may be responsible for dyslipidemia [26,29,30]. Several papers in the literature show that hyperandrogenemia might be associated with dyslipidemia, especially in obese people with polycystic ovary syndrome [31,32]. In our study, it was observed that patients with polycystic ovary syndrome had higher serum total testosterone levels than the control group. However, no correlation between serum total testosterone and lipid levels could be detected in our study. Furthermore, no significant correlation was observed between oncostatin M and total testosterone levels. Although some studies have examined the link between oncostatin M and serum total testosterone levels, the results of these studies are controversial and have primarily been conducted on rats [11,33]. Subsequently, this relationship should be scrutinized in future studies.

It has been revealed that patients with polycystic ovary syndrome have a high risk for diabetes mellitus, metabolic syndrome, and cardiovascular events. Significantly, the phenotype A has the highest risk for diabetes mellitus. Despite being treated, there was no reduction in the prevalence of metabolic syndrome and diabetes mellitus in patients with phenotype A of polycystic ovary syndrome [12]. Similarly, patients with polycystic ovary syndrome were observed to have high levels of fasting plasma glucose and homeostasis model assessment of insulin resistance in our study. It has also been reported that for phenotype-A polycystic ovary syndrome, the estimated average glucose level was predicted by decreased HDL-C and elevated free estradiol [34].

This study is one of the first to investigate the levels of oncostatin M in patients with polycystic ovary syndrome, filling a gap in the existing scientific literature on the close relationship between oncostatin M and human oocytes, granulosa cells, and ovulation [7,8]. However, the oncostatin M receptor and suppressor of cytokine signaling were not evaluated in this study. These factors could be helpful in understanding the underlying reason for ovulatory dysfunction in patients with polycystic ovary syndrome. It has been revealed that there is a strong connection between inflammation and polycystic ovary syndrome in which high levels of C-reactive protein are considered a sign of evoked inflammatory response, along with other inflammatory cytokines, such as IL-6 [13,14,15,16,17,18,19,20,21,22]. This study has a significant limitation as it did not assess the plasma levels of inflammatory cytokines.

## 5. Conclusions

Polycystic ovary syndrome is a complex disease characterized by biochemical or clinical hyperandrogenemia and metabolic disorders. In this research, the concentration of oncostatin M in the plasma was strikingly low in those with polycystic ovary syndrome, and this was significantly associated with the hormonal and metabolic features of polycystic ovary syndrome.

## Figures and Tables

**Figure 1 biomedicines-12-00355-f001:**
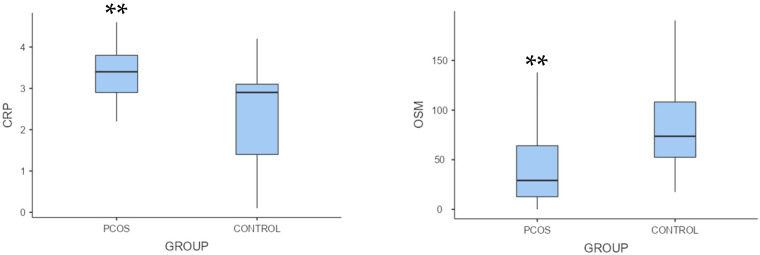
Box plots of oncostatin M and C-reactive protein levels in the control and polycystic ovary syndrome groups. Note. ** *p* < 0.01. PCOS, polycystic ovary syndrome; CRP, C-reactive protein; OSM, oncostatin M.

**Figure 2 biomedicines-12-00355-f002:**
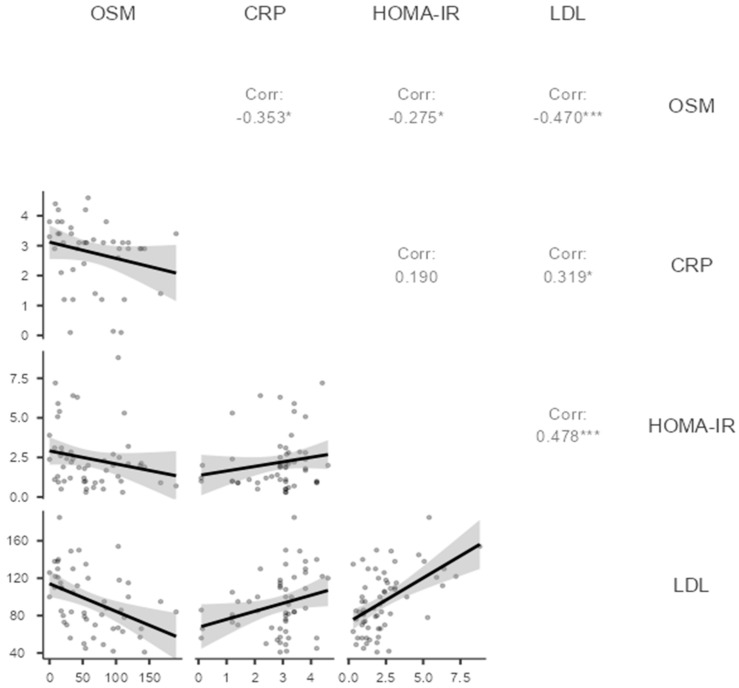
Scatter plots of oncostatin M with C-reactive protein, homeostasis model assessment of insulin resistance, and low-density lipoprotein in the polycystic ovary syndrome group. Note. * *p* < 0.05, *** *p* < 0.001. Spearman correlation analysis. OSM: Oncostatin M, CRP: C-reactive protein, HOMA-IR: Homeostasis model assessment of insulin resistance, LDL: low-density lipoprotein.

**Table 1 biomedicines-12-00355-t001:** The metabolic and hormonal features of the study groups.

Variables	PCOS (*n =* 32)	Control (*n =* 32)	*p*-Value
Age (years)	26.00 ± 4.67	25.00 ± 5.18	0.54
BMI (kg/m^2^)	24.95 ± 4.84	23.88 ± 2.82	0.63
Fasting glucose (mg/dL)	91.16 ± 6.02	84.19 ± 6.08	<0.001 ***
HOMA-IR	3.25 ± 1.95	1.42 ± 0.99	<0.001 ***
Total cholesterol (mg/dL)	193 ± 32.5	150 ± 24.7	<0.001 ***
LDL cholesterol (mg/dL)	115.56 ± 28.21	74.75 ± 22.12	<0.001 ***
Non-HDL cholesterol (mg/dL)	139.28 ± 35.11	95.78 ± 25.28	<0.001 ***
HDL (mg/dL)	53.88 ± 3.93	55.00 ± 7.59	0.51
LH/FSH ratio	1.73 ± 0.31	0.61 ± 0.20	<0.001 ***
Total testosterone (μg/L)	0.64± 0.21	0.40± 0.11	<0.001 ***
WBC	7859 ± 2299	7159 ± 1719	0.21
C-reactive protein (mg/dL)	4.37 ± 3.01	3.08 ± 4.14	0.001 **
Oncostatin M (pg/mL)	79.70 ± 107.23	166.11 ± 196.04	0.002 **

Note. ** *p* < 0.01, *** *p* < 0.001. BMI: body mass index; FSH: follicle-stimulating hormone; LH: luteinizing hormone; HOMA-IR: Homeostasis model assessment of insulin resistance; non-HDL: non-high-density lipoprotein; LDL: low-density lipoprotein; WBC: white blood cell.

**Table 2 biomedicines-12-00355-t002:** Correlation analysis of oncostatin M with other biochemical parameters.

Variables	PCOS Group Correlation Coefficient (ρ)	*p*-Value	Control Group Correlation Coefficient (ρ)	*p*-Value
Fasting glucose	−0.329	0.017 *	−0.230	0.213
HOMA-IR	−0.275	0.048 *	−0.130	0.486
Total cholesterol	−0.386	0.005 **	−0.130	0.470
LDL cholesterol	−0.470	<0.001 ***	−0.250	0.161
Non-HDL cholesterol	−0.440	0.001	−0.270	0.132
LH/FSH ratio	−0.316	0.023	−0.180	0.321
Total Testosterone	0.220	0.118	0.120	0.508
C-reactive protein	−0.353	0.019	−0.140	0.433

Note. ρ = Spearman correlation coefficient, * *p* < 0.05, ** *p* < 0.01, *** *p* < 0.001. PCOS, polycystic ovary syndrome; HOMA-IR: Homeostasis model assessment of insulin resistance, LDL: low-density lipoprotein; non-HDL: non-high-density lipoprotein; LH: luteinizing hormone; FSH: follicle-stimulating hormone.

## Data Availability

The datasets used and analyzed during the current study are available from the corresponding author upon reasonable request.

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
