# Peer review of "Oncostatin M Is Related to Polycystic Ovary Syndrome-Case Control Study"

_biomedicines, 2024, doi:10.3390/biomedicines12020355_

Round 1
Reviewer 1 Report
Comments and Suggestions for Authors
The authors tried to associate oncostatin M with the presence of a PCOS diagnosis.
1. The title is too strong. The authors are kindly requested to tone it down.
2. The sample is too small to extract any sound conclusions.
3. The authors are kindly requested to comply their manuscript according to STROBE Statement for case-control studies. They should report on each and every paragraph of the STROBE Statement.
4. Why did the authors use ROC curves? Again, the sample is too small.
5. The language of the paper needs thorough editing. Phrases such as "we.."need to be eliminated. Passive voice is preferable in scientific papers.
Comments on the Quality of English LanguageThe language of the paper needs thorough editing. Phrases such as "we..", or "our" need to be eliminated. Passive voice is preferable in scientific papers.
Author Response
- The title of the manuscript was toned down, as you requested.
- We added this to the limitation of this study. It was Dr. Figen Efe Camili's thesis study, and she funded it.
- STROBE checklist added.
- We used the ROC curves to represent the overall performance of oncostatin M to discriminate patients with PCOS from normal ones. Also, it was used to determine a cut-off value for oncostatin M. We could not find any literature regarding patient number limitations for plotting ROC curve. On the contrary, we provided references from literature in which ROC curves were used in low-patient settings.
- Deng J, Zhang W, Xu M, Liu X, Ren T, Li S, Sun Q, Xue C, Zhou J. Value of spectral CT parameters in predicting the efficacy of neoadjuvant chemotherapy for gastric cancer. Clin Radiol. 2024 Jan;79(1):51-59. doi: 10.1016/j.crad.2023.08.023. Epub 2023 Sep 20. PMID: 37914603.
- Li JL, Xu Y, Xiang YS, Wu P, Shen AJ, Wang PJ, Wang F. The Value of Amide Proton Transfer MRI in the Diagnosis of Malignant and Benign Urinary Bladder Lesions: Comparison With Diffusion-Weighted Imaging. J Magn Reson Imaging. 2024 Jan 4. doi: 10.1002/jmri.29199. Epub ahead of print. PMID: 38174777.
- The scientific language of the manuscript was thoroughly edited, and the phrases were replaced with passive ones.
Reviewer 2 Report
Comments and Suggestions for Authors
The manuscript “Oncostatin M is associated with the pathogenesis of Polycystic Ovary Syndrome and its metabolic complications” by Figen Efe Camili et al. analyzed oncostatin M in serum of controls and patients with polycystic ovary syndrome. They show that patients had lower levels.
High levels of oncostatin M have been detected in many chronic inflammatory conditions characterized by fibrosis, and this should be mentioned in the introduction.
Please correct “; adipokins“ “± 3,01“ „e and inflamatory signal mechanisms”
„A few papers in the literature have explored oncostatin M;” please include references.
How was serum diluted for oncostatin M analysis?
Correlation analysis has to be done in patients and controls separately.
“Based on this relationship, we can hypothesize that patients with polycystic ovary syndrome [10] may experience low-level inflammation” this is not correct because CRP of patients is induced
“Nonetheless, it is unclear whether the inflammation is caused by polycystic ovary syndrome or by insulin resistance and 200 obesity.” Patients have similar BMI as controls arguing against obesity.
Discussion has to be more focused on the current findings.
However, we did not analyze the relationship between serum total testosterone and lipid levels in our study because the patients with polycystic ovary syndrome were not obese” this can be nevertheless analyzed.
Serum IL-6 should be measured in patients and controls to identify an association with oncostatin M.
Comments on the Quality of English LanguageMinor corrections needed
Author Response
- The relationship of oncostatin M with fibrosis was added to the introduction section.
- We corrected “adipokines, “ “± 3,01, “ and “e and inflammatory signal mechanisms” in our text.
- References included in the text
- The serum is used without diluting.
- We added a correlation analysis for the control group.
- In light of the literature, we can conclude that patients with PCOS have chronic low-grade inflammation based on high CRP levels. We made a slight change in the meaning of the sentence
- Ref: Rudnicka E, Suchta K, Grymowicz M, Calik-Ksepka A, Smolarczyk K, Duszewska AM, Smolarczyk R, Meczekalski B. Chronic Low Grade Inflammation in Pathogenesis of PCOS. Int J Mol Sci. 2021 Apr 6;22(7):3789. doi: 10.3390/ijms22073789. PMID: 33917519; PMCID: PMC8038770)
- We changed the sentence to “Nonetheless, whether the inflammation is caused by polycystic ovary syndrome or insulin resistance is unclear. ”
- The discussion section was thoroughly revised and shortened based on our findings.
- We analyzed the relationship of testosterone with lipids and could not detect any relationship. We mentioned it in the discussion section but didn’t provide statistical data.
- We added this issue to the study's limitations.
Round 2
Reviewer 1 Report
Comments and Suggestions for Authors
The authors have revised their manuscript but there are still problems. My concerns:
1. The ROC Curve Analysis showed that oncostatin - M has only 50% of sensitivity and 75% of specificity.
2. The authors mention an effect size of 0.7. How did they reach to this number since the papers on oncostatin - M are scarce ? Also, It is practically certain that any random sample of a normal distribution will be skewed. This is a major limitation for ROC curve analysis.
Comments on the Quality of English LanguageNone
Author Response
- Thank you very much for taking the time to review this manuscript.
- After a heated discussion, we have decided to remove the ROC curves from the study based on your feedback.
- We used the open-source G*Power software for power analysis. We reached an effect size of 0.7 after using the published article by Nikanfar et al., in which the actual power was 0.99 in study groups for follicular fluid level of oncostatin M (n=30 per group).
Reviewer 2 Report
Comments and Suggestions for Authors
"Consider ing that there have been only a few scientific papers investigating the relationship between polycystic ovary syndrome and oncostatin M, this could be seen as a limitation of this study when trying to make a comparison." Please include the relevant references.
Legend Figure 2, please specify whether these correlations were done in the whole cohort or patients only.
Comments on the Quality of English LanguageEnglish is o.k. minor corrections are needed.
Author Response
- Thank you very much for taking the time to review this manuscript.
- We added the relevant references to the limitations of the study.
- We revised the legend in Figure 2.
Round 3
Reviewer 1 Report
Comments and Suggestions for Authors
No further comments.
Author Response
Thank you for reviewing the manuscript.
Best regards